# Brief communication: Nowcasting of precipitation for leading edge erosion-safe mode

Anna-Maria Tilg[1], Charlotte Bay Hasager[1], Hans-Jürgen Kirtzel[2], Poul Hummelshøj[3]

[1]Department of Wind Energy, Technical University of Denmark, Roskilde, 4000, Denmark
[2]METEK Meteorologische Messtechnik GmbH, Elmshorn, 25337, Germany
[3]METEK Nordic ApS, Roskilde, 4000, Denmark

*Correspondence to*: Anna-Maria Tilg (anmt@dtu.dk)

**Abstract.** Leading edge erosion (LEE) of wind turbine blades is caused by the impact of hydrometeors, which appear in solid or liquid phase. A reduction of the wind turbine blades' tip speed during defined precipitation events can mitigate LEE. To
apply such an erosion-safe mode, a precipitation nowcast is required. Theoretical considerations indicate that the time a raindrop needs to fall to the ground is sufficient to reduce the tip speed. Furthermore, it is described that a compact vertical pointing radar that measures rain in different heights with a sufficient high spatio-temporal resolution can nowcast rain for an erosion-safe mode.

## 1 Introduction

Leading edge erosion caused by precipitation impinging on blades with high tip speed results in rougher blades and a loss in annual energy production. According to Chen et al. (2019), the most expensive and most time-consuming process within maintenance of wind turbines is blade repair. Durable leading edge coatings are not yet available (Herring et al., 2019).

Bech et al. (2018) propose to reduce the tip speed during severe precipitation events to mitigate the effect of impacting
hydrometeors on the leading edge. They present five different erosion-safe modes where the tip speed of 90 m/s is reduced depending on the rain intensity (RI). For RI ≥ 20 mm/h the tip speed is reduced between 20 and 35 m/s, for RI ≥ 10 mm/h between 10 and 25 m/s and for RI ≥ 5 mm/h up to 20 m/s. These erosion-safe modes lead to an increase in the expected lifetime from 1.6 years up to 107 years assuming a specific rain climate. Furthermore, they investigate the influence of the turbine control during intense rain events on the annual energy production (AEP). The calculated AEP values range from negligible
reductions to significant increases. These calculations are based on the assumption that the time of reduced tip speed is 3 times longer than the actual time with RI above the mentioned thresholds. The suggested erosion-safe modes combine wind and precipitation measurements with a damage model. The damage model itself describes the erosion rate in relation to rain parameters (e.g. kinetic energy or accumulated amount of rain) and is based on laboratory measurements. Hasager et al. (2020) find higher erosion rates at coastal stations than inland stations in Denmark due to more intense rain events at high wind speeds

at these locations. Furthermore, they show an increase in the profit reducing the tip speed to 60 m/s or lower in case RI exceeds 1 mm/h.

The method of erosion-safe mode control is only possible to implement based on adequate precipitation nowcasting at minute to second scale. To limit the power production loss it is important to reduce the tip speed as early as possible, as long as needed

and as short as possible.

Nowcasting of rain characteristics for leading edge erosion-safe mode control based on radar and Doppler lidar is a brand new topic in wind energy. The proposed precipitation nowcasting for erosion-safe mode has similarity to short-term forecasting for power production based on ground-based remote sensing technologies like dual-Doppler radar (Valldecabres et al., 2018a) and

long-range scanning lidar at the minute scale (Valldecabres et al., 2018b). Also lidar-assisted yaw control, wake steering and induction control at the minute to second scale observed from turbine-mounted lidars (Würth et al., 2019) are comparable to precipitation nowcasting.

Radars are traditional instruments for precipitation observations while coherent Doppler lidar is novel in relation to rain (Aoki

et al., 2016; Sjöholm and Mikkelsen, 2018). This brief communication focuses on the radar-based precipitation nowcasting for erosion-safe mode.

## 2 Theory

The time until a hydrometeor hits the ground is governed by three parameters: distance between cloud base height and the ground and the type and size of the hydrometeor, which determine the resulting fall velocity.


The distance between the cloud base height and the ground depends mainly on the location, the storm type and the related cloud type. It can vary between a few hundreds to some thousands of meters. Depending on the storm type and the related growth mechanisms of cloud droplets, hydrometeors falling out of the cloud are liquid (drizzle, rain) or solid (snow, graupel, hail). Solid hydrometeors start to melt when they pass the 0°C isotherm, which is the upper boundary of the melting layer. In

weather radar measurements this layer is identified by high reflectivity values and is therefore called bright band. Thurai and Iguchi (2000) present a seasonal- and latitude-dependent distribution of the bright band height for stratiform events based on satellite measurements where the bright band height is the height with the highest reflectivity value. They find large seasonal variations of the bright band height for higher latitudes. Furthermore, they show that the 0°C isotherm from Recommendation ITU-R P.839-1 is usually 500 m or less above the bright band height. According to an updated version, P.839-4, the mean

annual 0°C isotherm is around 2000 m above sea level for Denmark (International Telecommunication Union, 2013). This distance leads to a bright band height of about 1500 m, which can be taken as a rough approximation for the distance a raindrop

falls until it reaches the ground. The bright band height in Denmark varies from about 3500 m in summer to 0 m in winter (Rashpal S. Gill, Danish Meteorological Institute, personal communication).

Rain consists of different raindrop sizes due to collision-induced breakup and coalescence of raindrops. In general, a single raindrop has a diameter between 0.1 mm and 8 mm, although raindrops with a diameter of 10 mm have been observed in relation to tropical clouds (Jones et al., 2010). Small drops up to around 1 mm are spherical, while larger drops have the shape of a flattened sphere. However, raindrops with a diameter above 6 mm are rare as they break up due to their flatten shape and the related hydrodynamic instability or due to collision with another raindrop. Bringi et al. (2003) compare raindrop size

distributions (DSD) from different climates. They find for convective storm types a mass-weighted mean diameter ($D_m$) between 1.50-1.75 mm for maritime-like environments and slightly larger $D_m$ between 2.00-2.75 mm for continental-like environments. For stratiform storms they report $D_m$ values between 1.25 and 1.75 mm but no clear distinction between different environments. These $D_m$ variations show that beside location dependent influences, raindrop formation processes related to specific storm types play a major role in determining the DSD.


Beside its shape, the fall velocity of a hydrometeor is controlled by three forces: gravity, buoyancy and the aerodynamic drag force. The fall velocity of a raindrop in still air, called terminal fall velocity, increases with the drop diameter. This velocity increase is approximately linear for small sizes and non-linear for large sizes. One of the most used empiric equations to calculate the fall velocity of raindrops is based on investigations from Atlas et al. (1973) . However, this equation does not

take into account the altitude dependence of the fall speed due to the reduced aerodynamic drag force with decreasing air density with increasing altitude. Jones et al. (2010) provide an equation considering a density ratio factor compared to the standard atmosphere to consider this altitude dependent change. Raindrops might not achieve terminal fall velocity during (heavy) rain, because the collision-induced breakup and coalescence of drops causes repetitive increase and decrease of the fall velocity (Jones et al., 2010). Furthermore, as rain consists of different drop sizes, there will be always raindrops that are

faster and slower.

Assuming a raindrop with a diameter of 1.5 mm, its terminal velocity is around 5 m/s taking the equation of Atlas et al. (1973). Considering a rain height of 1500 m, the raindrop needs 300 s (5 min) to fall to the ground. This time can be used to decelerate the tip speed of the wind turbine blades to reduce the impact energy by the drop and therefore the erosion of the leading edges.

For comparison, a larger raindrop with a diameter of 2.5 mm has a terminal velocity of around 7 m/s and needs 214 s (3.6 min) for the same distance.

Solid hydrometeors have different properties than raindrops. This difference results in different fall properties and impact behaviours on the leading edge of the wind turbine blade. The impact of hail and graupel causes more damage compared to

rain. The focus of this publication is on the nowcasting of rain, as hail and graupel are less frequent (Macdonald et al., 2016) and snow is not relevant.

## 3 Application

Operational nowcasting provided by national weather services, like Integrated Nowcasting through Comprehensive Analysis (Haiden et al., 2011), combines available observations from weather stations, weather radars and satellites with forecasts of

numerical weather prediction models. They provide values of precipitation amount and type beside other parameters in real time. However, in offshore environments, where enhanced leading edge erosion is observed, observations from weather stations are usually not available. National operated and usually onshore installed C- and S-band weather radars cover large areas, including many offshore wind farms, with a temporal resolution of $\geq$ 5 minutes. However, some notable disadvantages of these weather radars for a nowcast of precipitation for offshore wind farms are:

- Partial beam filling: The precipitation does not fill completely the scanned volume, because it increases with increasing distance from the radar. This condition can lead to an underestimation of RI.

- Overshooting: The height of the radar beam is above the precipitation, because the height of the radar beam increases due to the scan elevation angle and the curvature of the Earth. This condition can lead to an underestimation of RI or even failure to detect precipitation.

- Clutter caused by wind farms: Reflections produced by wind farm infrastructure indicate wrongly precipitation.

These and other limitations like anomalous propagation of the radar beam can be detected but for some cases a correction is difficult (e.g. beam filling). This situation leads to some uncertainty in the precipitation parameters. Local installed sensors measuring vertical profiles of precipitation are therefore an interesting option for nowcasting using the described time difference between the detection and impact of raindrops. Takahashi (1990) presents the Precipitation Particle Image Sensor

(PPIS). This sensor measures like a radiosonde the precipitation at a certain height while ascending through the atmosphere. In contrast, vertical pointing radars provide continuous precipitation measurements in different altitudes at the same time.

An example for a ground-based vertical pointing radar is the Micro Rain Radar (MRR) from METEK. It is a compact 24 GHz (K-band) frequency modulated continuous wave (FM-CW) Doppler radar with a parabola antenna pointing vertically (Peters

et al., 2002). The latest model MRR-PRO has a vertical resolution of > 10 m and can provide an averaged Doppler spectrum of the hydrometeors in $\geq$ 1 s, i.e. a Doppler spectrum roughly each 10 m for each second is available.

In case of rain, the first moment of the measured Doppler spectra allows the estimation of the fall velocity of the raindrops via the Doppler velocity. Based on the calculated fall velocity, the raindrop size can be estimated using the previously mentioned

relation between these two parameters inversely. The availability of the raindrop size and fall velocity allows the calculation

of further rain parameters like the rain reflectivity and RI (assuming Rayleigh approximation) for different heights. These calculations assume that only raindrops and no solid hydrometeors or mix of both (i.e. sleet) backscatter the signal.

Figure 1 shows the temporal and vertical evolution of the radar reflectivity, fall velocity and RI of an event in December 2019 in Plymouth (United Kingdom) measured with a MRR-PRO. This MRR-PRO provided data every 10 s up to 3200 m above ground. The high values of the derived parameters reflectivity and RI between 2000 m and 1600 m indicate a melting layer. Below this layer, precipitation falls as rain, where RI close to the ground is above 5 mm/h for several consecutive minutes. Rain that was registered at the lower boundary of the melting layer for example at 17:34 arrived around 2 min later at turbine hub height (approximately 100 m above ground). This time difference is shorter than the expected time based on the above calculations. One reason is the reduced air density and therefore reduced aerodynamic drag in higher altitudes that leads to a higher fall velocities. Additionally, because of break-up and coalescence processes, the actual fall velocity can differ from the terminal fall velocity with velocities even above terminal fall velocity (Montero-Martínez et al., 2009). Nevertheless, in principle the measured time difference would enable the erosion-safe mode control to reduce the tip speed of the wind turbine blades in due time when observing a rain event with light or moderate RI.

Although the tip speed has maybe already been reduced when observing heavy or violent rain events (RI > 10 mm/h), following the suggested RI thresholds from Bech et al. (2018) or Hasager et al. (2020) for applying an erosion-safe mode, it is still important to measure events with such a high RI. Adirosi et al. (2016) observe an increase of the median volume diameter of raindrops from 1.25 mm at 1050 m AGL to 2.07 mm at 105 mm AGL during the convection phase of a rain event with high RI. This increase is probably due to coalescence and drop sorting. Therefore, it is possible that the RI at the wind turbine is higher than measured at some distance for the nowcast. The nowcast would not be so effective, except measurements closer to the wind turbine would be included to check for such an increase. As larger drops fall faster, the time for reducing the tip speed in due time is shorter.

Nevertheless, an advantage of a MRR is that the height information of the melting layer can also help to identify the risk of blade icing, especially in cold climates. Furthermore, the MRR measurements are not disturbed by the flow around the sensor in contrast to in-situ sensors like disdrometers (Testik and Rahman, 2016). However, in events with notable vertical wind (e.g. thunderstorms) the calculated RI based on the MRR-PRO raw data includes some error as still air is assumed. The radar beam of the MRR-PRO is attenuated stronger in upper heights (> 1 km) during violent RI compared to C- or S-band radar beams. The parameter Path Integrated Attenuation (PIA) of the MRR-PRO contains this information and can help to identify violent rain events.

The automatic detection of solid hydrometeors with an MRR is still challenging as these precipitation types have different fall properties than rain. However, they can be detected by the synopsis of different rain parameters provided by the MRR.

## 4 Conclusion

Erosion-safe mode needs, like other parameters in wind turbine controlling, a nowcasting with high temporal and spatial resolution. Theoretical investigations showed that it takes a raindrop around five minutes (or less) to cover the distance between the melting layer and the ground. If the raindrop is detected when it starts to fall, this time difference is sufficient to enable erosion-safe mode with reduced tip speed. Vertical precipitation profiles can be obtained using vertical pointing radars. For example, the Micro Rain Radar (MRR) from METEK points strictly vertically and measures Doppler spectra up to three kilometres with a resolution > 10 m. Due to the high temporal resolution, the Doppler spectra and the related rain parameters are updated frequently and can be used for nowcasting. Using a vertical pointing radar also allows capturing the height and temporal evolution of a possible present melting layer and solid hydrometeors. Based on these reflections it is possible to measure and nowcast rain where vertical precipitation profiles with a high spatio-temporal resolution are essential. This nowcasting technique can be applied onshore and offshore. Future work includes the combination of vertical pointing radar measurements and damage models to improve erosion-safe mode models and their operational use.

**Author contribution**

**A.-M. Tilg:** developed and discussed the concept, wrote main parts of the text. **C. Hasager:** discussed concept, wrote and edited text. **H.-J. Kirtzel:** discussed the concept, edited text. **P. Hummelshøj:** discussed the concept, edited text.

**Competing interests**

The author **H.-J. Kirtzel** is employed by the private company METEK GmbH and author **P. Hummelshøj** is employed by the private company METEK Nordic ApS. The companies develop, produce and sell the Micro Rain Radar (MRR). The authors declare that they have no other known competing financial interests or personal relationships that could have appeared to influence the work reported in this paper.

**Acknowledgements**

We thank Chris Kidd (University of Maryland, NASA) for providing us the measurement example of the Micro Rain Radar installed at the Plymouth Marine Laboratory where Tim Smyth is responsible for the observations. This work was supported by the Innovation Fund Denmark grant 6154-00018B for the project EROSION (http://www.rain-erosion.dk; last access 13 January 2020). We thank two anonymous reviewers for their comments.

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

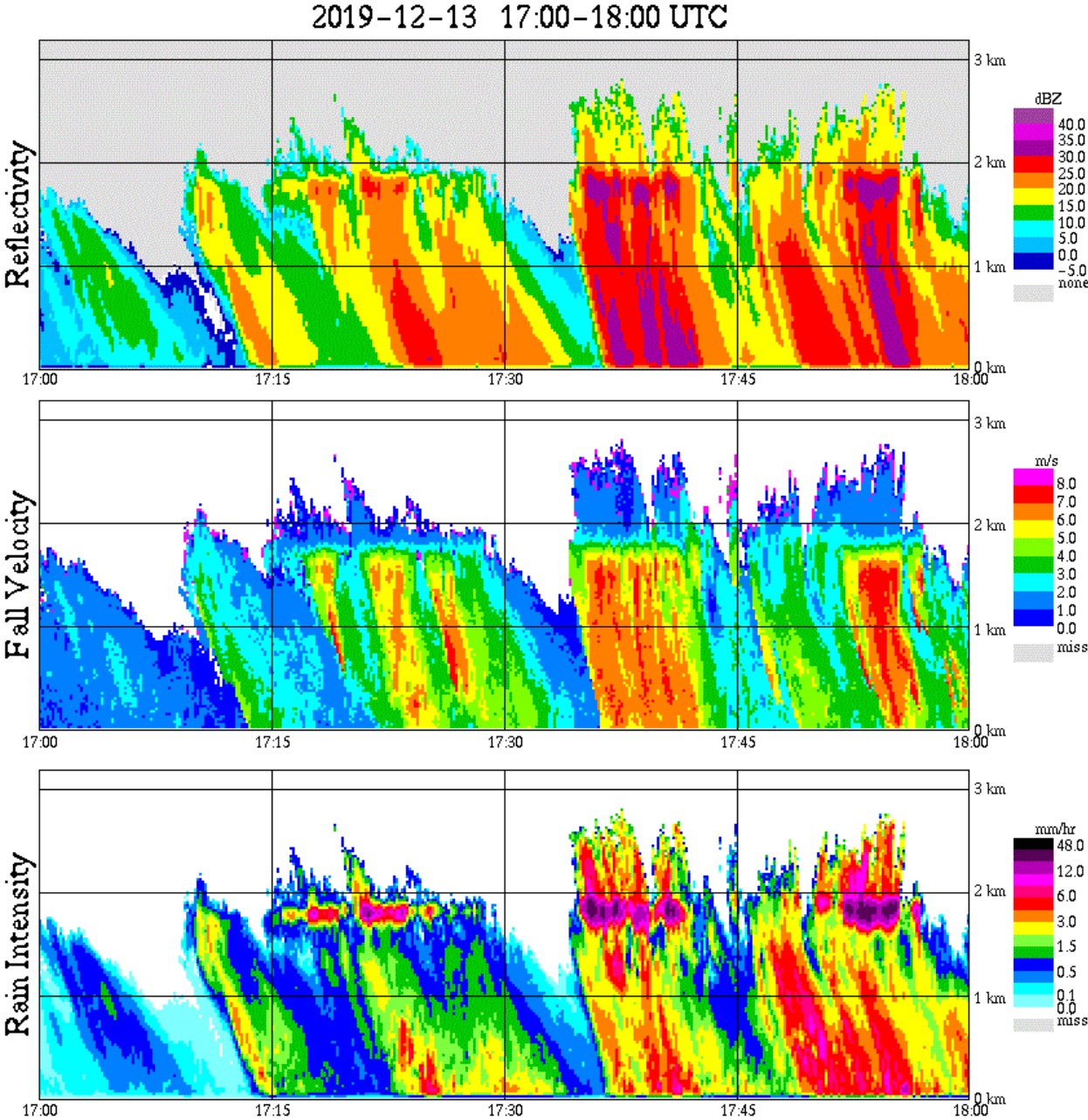

**Figure 1:** Radar reflectivity [dBz], fall velocity of raindrops [m/s] and rain intensity [mm/h] based on Micro Rain Radar measurements in Plymouth (United Kingdom). The vertical axis describes the vertical distance from the sensor and the horizontal axis the time in UTC.