# Peer review of "Brief communication: Nowcasting of precipitation for leading edge erosion-safe mode"

_Wind Energy Science, 2020_

## Referee Comment (RC1) · Anonymous Referee #1 · 29 Mar 2020

General Comments:

The authors present an interesting concept that aims to anticipate severe precipitation events. This can be used to inform an erosion-safe mode and reduce the tip speed of a turbine before the event impacts. The concept addresses the limitations of using forecasting approaches. Overall, I believe this paper would be suitable for publication if the following points are addressed.

The study builds on work from Bech, 2018 and Hasager, 2020. I would consider including a paragraph introducing their work and findings so that the reader can understand how this work fits into the wider research.

The authors need to consider the worst case scenario in this study. If high intensity

events cause the most erosion damage, then how the nowcasting would respond to an extreme intensity event needs to be presented. If an extreme, highly damaging event does not provide enough time for the turbine to be slowed than there is a limit on the application of the erosion-safe mode. The concept has been presented successfully for a moderate intensity event.
* * *
Specific Comments:

1. Page 1, sentence starting line 18, "Therefore, the method...".

I do not think that you should say the erosion safe mode is a "solution". As far as I'm aware, the erosion-safe mode has not been proven on an in-operation wind turbine. It would be more appropriate to present the safe mode as a "theory" e.g. "Bech proposes the idea of reducing tip speeds..."

2. Page 2, sentence starting line 54, "Raindrops have diameters...".

A reference to a study where 8mm and 10mm droplet diameters have been recorded is required. Alternatively, data validating this sentence should be provided.

3. Page 3, paragraph starting line 73, "Assuming a raindrop with...".

How does the fall time change when you take a worst case scenario (i.e. largest recorded droplet size, minimum aerodynamic drag from the altitude, etc)? If the erosion-safe mode is only intended to be used in severe precipitation events, the severe case needs to be presented.

4. Page 4, paragraph starting line 108, "Figure 1 shows the..."

Can you show a radar reflectivity for a high intensity event? You present a plot for an intensity of 5 mm/h and calculate that rain arrives at the ground after two minutes. As with the previous comment, the worst-case scenario needs to be presented. How would this differ for an extreme intensity of e.g. 100mm/h? Is it still possible to reduce

the tip speed of the turbine before the event arrives? Also, you should consider the height of a turbine in your calculation, rather than time to the ground.

5. Page 4, sentence starting line 123, "The MRR measurements are...".

Can you provide a reference to flow disturbance around disdrometers?

6. Page 6, sentence starting line 145, "This nowcasting technique...".

You mention that the technique can be applied offshore. However, earlier in the paper you discuss uncertainties in offshore radars, which indicates there are challenges to using this offshore. Can you extrapolate on this point to clear up the confusion?
* * *
Technical Corrections:

1. Page 3, sentence starting line 63, "The fall velocity of a..."

Change to: "Besides its shape, the fall velocity of a hydrometeor is controlled by three forces:..."

2. Page 3, sentence starting line 66, "This increase is linearly..."

Replace "linearly" with "linear"

3. Page 6, line 136.

Remove extra line after "4 Conclusion'"
* * *
Thank you for considering my comments. I found this work interesting and believe it could help in addressing leading edge erosion.

---

## Referee Comment (RC2) · Anonymous Referee #2 · 13 Apr 2020

The manuscript is well written and presents its idea and concept in a clear and concise fashion.

It could be beneficial to present a study indicating the loss of AEP of the turbine when using a controller of this kind to decrease the blade tip velocity vs the loss of AEP and expected costs for repair of erosion for a current case with no nowcasting controller. This way, the financial benefits would be clearer.

---

## Author Comment (AC1) · 11 May 2020

Please find our response to the comments of the reviewers as well as the tracked-chages version of the manuscript in the attached document.

Please also note the supplement to this comment:
https://www.wind-energ-sci-discuss.net/wes-2020-4/wes-2020-4-AC1-supplement.pdf

---

## Author Response (AR1)

**General Comments:**

**The authors present an interesting concept that aims to anticipate severe precipitation events. This can be used to inform an erosion-safe mode and reduce the tip speed of a turbine before the event impacts. The concept addresses the limitations of using forecasting approaches. Overall, I believe this paper would be suitable for publication if the following points are addressed.**

Thank you for your positive feedback.

**(1)**
**The study builds on work from Bech, 2018 and Hasager, 2020. I would consider including a paragraph introducing their work and findings so that the reader can understand how this work fits into the wider research.**
Thank you for pointing to this shortcoming in the manuscript. We added following text to the introduction (line 19 to 27):

*Bech et al. (2018) propose to reduce the tip speed during severe precipitation events to limit erosion and to extend blade lifetime. For example, they investigate the influence of turbine control during intense rain events on the annual energy production (AEP). Based on some assumptions, e.g. the precipitation climate, they find that AEP increases when reducing the tip speed during intense rain events compared to no reduction. The suggested turbine control combines wind and precipitation measurements with a damage model. The damage model itself describes the erosion rate in relation to rain parameters (e.g. kinetic energy or accumulated amount of rain) and is based on laboratory measurements. Hasager et al. (2020) find higher erosion rates at coastal stations than inland stations in Denmark due to more intense rain events at high wind speeds at these locations. Furthermore, they show an increase in the profit reducing the tip speed to 60 m/s or lower in case the rain intensity exceeds 1 mm/h.*

**(2)**
**The authors need to consider the worst case scenario in this study. If high intensity events cause the most erosion damage, then how the nowcasting would respond to an extreme intensity event needs to be presented. If an extreme, highly damaging event does not provide enough time for the turbine**

**to be slowed than there is a limit on the application of the erosion-safe mode. The concept has been presented successfully for a moderate intensity event.**
Thank you for bringing up this point. Erosion-safe mode depends mainly on two parameters: (i) precipitation load, which is usually described with rain intensity and (ii) damage model, which depends on the material properties. The combination of these two parameters includes various scenarios, for example blade material that erodes fast during light rain intensities and other material that resists longer experiencing harsher conditions. Therefore, from our point of view it is difficult to define a general applicable worst-case scenario. Hasager et al. (2020) also considers the economical aspect of an erosion-safe mode and shows an increase in profit in case the tip speed is reduced at rain intensities of 1 mm/h or higher. Hence, the tip speed is already reduced when heavy rain (rain intensity ≥ 10 mm/h) is observed.

Considering a severe scenario, one needs also to consider the impact of hail and graupel. This type of precipitation occurs rarely but can cause more damage than intense rain events because of their impact behavior. As mentioned in the manuscript (line 132 to 133) it is possible to detect this type with a vertical pointing radar like the MRR from METEK.

To clarify this point in the manuscript, we added following sentence (line 123 to 124):

*Based on the results from Hasager et al. (2020) one can assume that at this rain intensity the tip speed will be reduced around this value.*
* * *
**Specific Comments:**

**(3)**
**1. Page 1, sentence starting line 18, "Therefore, the method...".**
**I do not think that you should say the erosion safe mode is a "solution". As far as I'm aware, the erosion-safe mode has not been proven on an in-operation wind turbine. It would be more appropriate to present the safe mode as a "theory" e.g. "Bech proposes the idea of reducing tip speeds..."**
We agree that erosion-safe mode should be described as possible solution as it has not yet been verified in a full-scale experiment. We reformulated the relevant part in the manuscript to (line 19 to 20):

*Bech et al. (2018) propose to reduce the tip speed during severe precipitation events to limit erosion and to extend blade lifetime.*

**(4)**
**2. Page 2, sentence starting line 54, "Raindrops have diameters...".**
**A reference to a study where 8mm and 10mm droplet diameters have been recorded is required. Alternatively, data validating this sentence should be provided.**
We added a relevant publication (line 61 to 62):

*Raindrops have diameters up to 8 mm, although raindrops with 10 mm have been observed in tropical areas (Jones et al., 2010).*

**(5)**
**3. Page 3, paragraph starting line 73, "Assuming a raindrop with...".**
**How does the fall time change when you take a worst case scenario (i.e. largest recorded droplet size, minimum aerodynamic drag from the altitude, etc)? If the erosion-safe mode is only intended to be used in severe precipitation events, the severe case needs to be presented.**
As mentioned in point (2) it is difficult to define a general applicable worst-case scenario. Furthermore, it is shown that profit is already gained by reducing the tip speed at rain intensities around 1 mm/h. As our intention is to show frequent occurring rain conditions, we do not see a need to re-calculate the theoretical fall time for more extreme values.

**(6)**
**4. Page 4, paragraph starting line 108, "Figure 1 shows the..." Can you show a radar reflectivity for a high intensity event? You present a plot for an intensity of 5 mm/h and calculate that rain arrives at the ground after two minutes. As with the previous comment, the worst-case scenario needs to be presented. How would this differ for an extreme intensity of e.g. 100mm/h? Is it still possible to reduce the tip speed of the turbine before the event arrives? Also, you should consider the height of a turbine in your calculation, rather than time to the ground.**
As mentioned in point (2) our assumption is that the tip speed of the wind turbine blades is already reduced at lower rain intensities. This consideration is based on theoretical calculations presented in Hasager et al. (2020). Furthermore, Herring et al. (2020) mention "… that erosion damage is not driven solely by heavy and violent precipitation …". Therefore, not only rain events with a rain intensity ≥ 20 mm/h are interesting, but also events with a moderate rain intensity, which occur more often.

Good point about taking turbine height instead of ground to determine the fall time. However, calculations show that a drop with 1.5 mm diameter and 5 m/s fall velocity needs 320 s from 1600 m to ground and 300 s from 1600 m to 100 m hub height. This is 6.25% reduced fall time. From our point of view this value does not significantly influence the general conclusion. Nevertheless, we changed the formulation to (line 117 to 118):

*Rain that was registered at the lower boundary of the melting layer for example at 17:34 arrived around 2 min later at turbine hub height (approximately 100 m above ground).*

We added following text about the AEP to the introduction (line 19 to 22):

[revised manuscript text omitted]

---

## Author Response (AR2)

We would like to thank the associate editor for the feedback. We have carefully considered every comment. Below we provide the detailed answers to the comments – in bold black the comments from the editor, in black our replies to the comments and in italic black passages from the updated manuscript.

**Associate Editor Decision**

**Comments to the Author:**
**Please elaborate on these points in the manuscript based on the text in response to review. This text inserted is insufficient. You have inserted non-quantitative sentences when you could be both specific and quantitative (as you have been in the response). Thank you.**

1) **Based on the results from Hasager et al. (2020) one can assume that at this rain intensity the tip speed will be reduced around this value. (This sentence needs work, the meaning is unclear, use quantitative responses)**

We agree that the formulation of this sentence is not good. As we included some information about the use of the presented nowcast during heavy rain, we included our intended message of this sentence in the new paragraph (see point 4 for details).

2) **Bech et al. (2018) propose reduce the tip speed during severe precipitation events to limit erosion and to extend blade lifetime. (by how much, to what effect, see also response to 2nd reviewer).**

We included the proposed rain intensity thresholds for active erosion-safe mode control and the related tip speed reductions. Furthermore, the influence on the expected lifetime and the AEP are described now in more detail (line 19 to 26).

*Bech et al. (2018) propose to reduce the tip speed during severe precipitation events to mitigate the effect of impacting hydrometeors on the leading edge. They present five different erosion-safe modes where the tip speed of 90 m/s is reduced depending on the rain intensity (RI). For RI ≥ 20 mm/h the tip speed is reduced between 20 and 35 m/s, for RI ≥ 10 mm/h between 10 and 25 m/s and for RI ≥ 5 mm/h up to 20 m/s. These erosion-safe modes lead to an increase in the expected lifetime from 1.6 years up to 107 years assuming a specific rain climate. Furthermore, they investigate the influence of the turbine control during intense rain events on the annual energy production (AEP). The calculated AEP values range from negligible reductions to significant increases. These calculations are based on the assumption that the time of reduced tip speed is 3 times longer than the actual time with RI above the mentioned thresholds.*

3) **Raindrops have diameters up to 8 mm, although raindrops with 10 mm have been observed in tropical areas (Jones et al., 2010). (Not detailed enough).**

We included more details about possible sizes of raindrops (line 65 to 69):

*In general, a single raindrop has a diameter between 0.1 mm and 8 mm, although raindrops with a diameter of 10 mm have been observed in relation to tropical clouds (Jones et al., 2010). Small drops up to around 1 mm are spherical, while larger drops*

have the shape of a flattened sphere. However, raindrops with a diameter above 6 mm are rare as they break up due to their flatten shape and the related hydrodynamic instability or due to collision with another raindrop.

4) **As mentioned in point (2) it is difficult to define a general applicable worst-case scenario. Furthermore, it is shown that profit is already gained by reducing the tip speed at rain intensities around 1 mm/h. As our intention is to show frequent occurring rain conditions, we do not see a need to re-calculate the theoretical fall time for more extreme values. (Not quantitative/detailed enough 1mm/hr is light rainfall. Its reasonable to ask what happends in heavy rainfall, please be specific about dropsize/amounts)**

Our assumption was that the tip speed is already reduced at rain intensities of 1 mm/h, as Hasager et al. (2020) showed an increase in profit using this threshold. We agree that this assumption was not clearly mentioned and might not be satisfactory to all readers. Therefore, we included following information (line 90 to 91):

*For comparison, a larger raindrop with a diameter of 2.5 mm has a terminal velocity of around 7 m/s and needs 214 s (3.6 min) for the same distance.*

Furthermore, we wrote following paragraph mentioning an observed increase of drop size towards ground during a convective event (resulting in a higher rain intensity) and possible consequences for the nowcast (line 141 to 148).

*Although the tip speed has maybe already been reduced when observing heavy or violent rain events (RI > 10 mm/h), following the suggested RI thresholds from Bech et al. (2018) or Hasager et al. (2020) for applying an erosion-safe mode, it is still important to measure events with such a high RI. Adirosi et al. (2016) observe an increase of the median volume diameter of raindrops from 1.25 mm at 1050 m AGL to 2.07 mm at 105 mm AGL during the convection phase of a rain event with high RI. This increase is probably due to coalescence and drop sorting. Therefore, it is possible that the RI at the wind turbine is higher than measured at some distance for the nowcast. The nowcast would not be so effective, except measurements closer to the wind turbine would be included to check for such an increase. As larger drops fall faster, the time for reducing the tip speed in due time is shorter.*

We also tried to make the pros and cons of an MRR-PRO for the nowcast clearer (line 150 to 156):

*Nevertheless, an advantage of a MRR is that the height information of the melting layer can also help to identify the risk of blade icing, especially in cold climates. Furthermore, the MRR measurements are not disturbed by the flow around the sensor in contrast to in-situ sensors like disdrometers (Testik and Rahman, 2016). However, in events with notable vertical wind (e.g. thunderstorms), the calculated RI based on the MRR-PRO raw data includes some error as still air is assumed. The radar beam of the MRR-PRO is attenuated stronger in upper heights (> 1 km) during violent RI compared to C- or S-band radar beams. The parameter Path Integrated Attenuation (PIA) of the MRR-PRO contains this information and can help to identify violent rain events.*

**5) Furthermore, the use of C- and S-band based weather radars, which are usually installed onshore, includes some limitations, e.g. precipitation does not fill completely the scanned volume, height of radar beam is above precipitation and reflections caused by the wind farm infrastructure wrongly indicate precipitation. These limitations can be corrected only to some extend and lead to some uncertainty in the precipitation parameters. (Check for typos, expand using the detailed text in response to reviewers.)**

As suggested we included the detailed text provided in the answer to the reviewer in the manuscript (line 102 to 112).

*National operated and usually onshore installed C- and S-band weather radars cover large areas, including many offshore wind farms, with a temporal resolution of ≥ 5 minutes. However, some notable disadvantages of these weather radars for a nowcast of precipitation for offshore wind farms are:*
- *Partial beam filling: The precipitation does not fill completely the scanned volume, because it increases with increasing distance from the radar. This condition can lead to an underestimation of RI.*
- *Overshooting: The height of the radar beam is above the precipitation, because the height of the radar beam increases due to the scan elevation angle and the curvature of the Earth. This condition can lead to an underestimation of RI or even failure to detect precipitation.*
- *Clutter caused by wind farms: Reflections produced by wind farm infrastructure indicate wrongly precipitation.*

*These and other limitations like anomalous propagation of the radar beam can be detected but for some cases a correction is difficult (e.g. beam filling). This situation leads to some uncertainty in the precipitation parameters.*